**Subject Area:**
genomics/systems biology

gene expression, expression plasticity, *cis*-element, transcription factor, histone modification

**Authors for correspondence:**
Fei He
e-mail: feihe@ksu.edu
Zhuo Du
e-mail: zdu@genetics.ac.cn

[†]These authors contributed equally to this study.

[‡]Present Address: Kansas State University, Manhattan, KS 66506, USA.

# Multivariable regulation of gene expression plasticity in metazoans

Long Xiao[1,2,†], Zhiguang Zhao[1,2,†], Fei He[3,‡] and Zhuo Du[1,2]

[1]State Key Laboratory of Molecular Developmental Biology, Institute of Genetics and Developmental Biology, Chinese Academy of Sciences, Beijing 100101, People's Republic of China
[2]University of Chinese Academy of Sciences, Beijing 10049, People's Republic of China
[3]Biology Department, Brookhaven National Lab, Upton, NY 11967, USA

FH, 0000-0002-1165-3248; ZD, 0000-0002-6322-4656

An important capacity of genes is the rapid change of expression levels to cope with the environment, known as expression responsiveness or plasticity. Elucidating the genomic mechanisms determining expression plasticity is critical for understanding the molecular basis of phenotypic plasticity, fitness and adaptation. In this study, we systematically quantified gene expression plasticity in four metazoan species by integrating changes of expression levels under a large number of genetic and environmental conditions. From this, we demonstrated that expression plasticity measures a distinct feature of gene expression that is orthogonal to other well-studied features, including gene expression level and tissue specificity/broadness. Expression plasticity is conserved across species with important physiological implications. The magnitude of expression plasticity is highly correlated with gene function and genes with high plasticity are implicated in disease susceptibility. Genome-wide analysis identified many conserved promoter *cis*-elements, *trans*-acting factors (such as CTCF), and gene body histone modifications (H3K36me3, H3K79me2 and H4K20me1) that are significantly associated with expression plasticity. Analysis of expression changes in perturbation experiments further validated a causal role of specific transcription factors and histone modifications. Collectively, this work reveals the general properties, physiological implications and multivariable regulation of gene expression plasticity in metazoans, extending the mechanistic understanding of gene regulation.

## 1. Introduction

Gene expression connects genotypes to phenotypes. Gene expression plasticity (GEP), which concerns the capacity of genes to change their expression levels under diverse conditions, is critical for phenotypic plasticity, adaptation and evolvability [1–4]. It has been widely observed that the expression levels of certain genes (such as stress-response genes) are intrinsically more flexible while those of other genes are more resistant (such as housekeeping genes). GEP has important implications for organismal fitness [2]. For example, the ability of genes to rapidly tune their expression levels to accommodate changing conditions (such as stress) is crucial for the organism to adapt to a new environment, hence increasing the fitness. Theoretically, low plasticity confers cellular stability and allows the organism to maintain a steady state, while high plasticity allows an organism to rapidly remodel its gene expression programmes and cellular function to cope with changing environments, enabling phenotypic adaption. Elucidating the genomic mechanisms underlying differential GEP is an important but unresolved question in genome biology.

Pioneering studies in yeast have defined GEP by measuring the magnitude of gene expression changes across diverse genetic and environmental conditions [3,5,6]. Functional analyses have revealed that two types of genetic

and epigenetic signatures in promoters correlate with expression plasticity. The first signature is the TATA box, a conserved element present in many eukaryotic gene promoters. In yeast, TATA box-containing genes exhibit significantly higher levels of plasticity than TATA-less genes, and this phenomenon is conserved in other species [3]. The second signature is nucleosome occupancy and organization near the transcription start site (TSS). The presence of well-positioned nucleosomes is associated with significantly higher expression plasticity in yeast [5,6].

In multicellular organisms, GEP and its regulation are less well understood. Gene expression changes have been profiled in response to a limited number of conditions in individual experiments in *Caenorhabditis elegans* [7–9], *Drosophila* [10] and human cells [11]. The presence of specific transcription regulatory elements has been shown to allow dozens of genes to be co-regulated in response to heat shock stress [7]. Systematic comparison of gene expression changes in five *C. elegans* strains cultured under five conditions identified that certain genes are more prone to respond to *trans*-acting factors to mediate genotype–environment interactions, especially those with complex promoter architecture and mid-range expression levels [8]. In addition, the genetical genomics approach has been employed to identify *cis*- and *trans*-loci that are strongly associated with gene expression changes in response to specific environmental conditions such as ambient temperature [9]. These studies have characterized groups of genes that are prone to change expression under specific stress conditions, and revealed certain gene features that mediate plastic gene expression. However, many important questions remain to be addressed. For example, is GEP an intrinsic gene property conserved in multicellular organisms? What are the biological implications of differential expression plasticity? Furthermore, given the significantly increased complexity of gene regulation in metazoan species, are there additional *cis*-elements, *trans*-factors and epigenetic regulators that underlie expression plasticity? To this end, a systematic analysis of the properties and genomic regulation of GEP in metazoan species is highly desirable, but has yet to be performed.

In this study, we performed an integrative functional analysis of genome-wide gene expression programmes, gene attributes, *cis*-regulatory motifs, *trans*-acting proteins and histone modifications to decipher the properties, implications and regulation of GEP in four metazoan species. Our results revealed that GEP is a conserved gene property implicated in cellular flexibility, disease susceptibility and stress/environmental response. We provide genome-wide evidence that core promoter *cis*-elements, transcription factors (such as CTCF) and gene body histone modifications (H3K36me3, H3K79me2 and H4K20me1) play a causal role in determining expression plasticity. Together, our findings provide insights into the function and regulation of GEP in metazoans.

## 2. Results

### 2.1. Quantification of gene expression plasticity in metazoan species

GEP is defined as the magnitude of gene expression change across diverse genetic and environmental conditions (figure 1a). Using a similar method as done previously for yeast data [3], we collected gene expression datasets from a public database [12] or from the literature [13] to quantify GEP in four metazoan species (electronic supplementary material, table S1, and Methods). For each condition (e.g. culture temperature, gene mutation, drug treatment), the magnitude of gene expression change after treatment was quantified as the square of log2-fold change, and the values across all conditions (range from 270 to 1267 in different species, figure 1b) were averaged and log-transformed to represent GEP (electronic supplementary material, figure S1a).

The measured GEP shows a wide distribution which is distinct from that expected, in which the fold changes of genes associated with each condition were randomized (figure 1c; electronic supplementary material, figure S1b). We performed a series of quality control checks to ensure reliable quantification of GEP. First, we verified that the total number of conditions used was sufficient to represent GEP. Simulation of the influence of condition number on GEP showed high stability of the score ($r = 0.9$) once the condition number reached 100 (electronic supplementary material, figure S1c); as we used over 100 conditions for all species, our condition number was sufficient. Second, we confirmed the biological relevance of quantified GEP using benchmark genes. As expected, human signal-responsive genes [14] whose expression is expected to be conditional and dynamic showed significantly higher GEP than other genes (electronic supplementary material, figure S1d and table S2). In addition, consistent with the expectation that genes participating in the stress response should have high GEP, our data showed that, in both fly and worm [10,15], stress-responsive genes and those required for stress response indeed exhibited significantly higher GEP (electronic supplementary material, figure S1e,f and table S2). Third, our measurement of GEP is based on quantifying expression changes across both normal and challenging conditions (such as environmental changes and genetic perturbations), which would better approximate the ability of a gene to change its expression than the natural variability of gene expression across tissue or cell types under normal condition. We applied the same method and recalculated GEP of human genes using expression data across a large collection of normal samples [16]. While GEP calculated using both data sources (termed $GEP_{N+C}$ and $GEP_N$, respectively) correlated moderately with each other (Spearman correlation, $\rho = 0.317$, $p < 0.001$, figure 1d), $GEP_{N+C}$ used here better represented expression plasticity than $GEP_N$. Specifically, $GEP_{N+C}$ exhibited a significantly higher level than $GEP_N$ (Wilcoxon signed-rank test, $p < 0.001$, figure 1e; electronic supplementary material, table S3) for signal-responsive genes that should exhibit high expression plasticity. Finally, unlike yeast in which GEP is quantified in one cell across various conditions, GEP in multicellular organisms combines the results of different tissues and cell types. Thus, the quantifying organismal-level GEP may not be meaningful if cell-specific GEP is pervasive. To assess this possibility, we calculated GEP for specific human cell lines and found that cell-type-specific GEP values correlated significantly with total GEP and with each other (figure 1f), suggesting that, while GEP exhibits cell specificity, the total GEP recapitulates that of specific cell types.

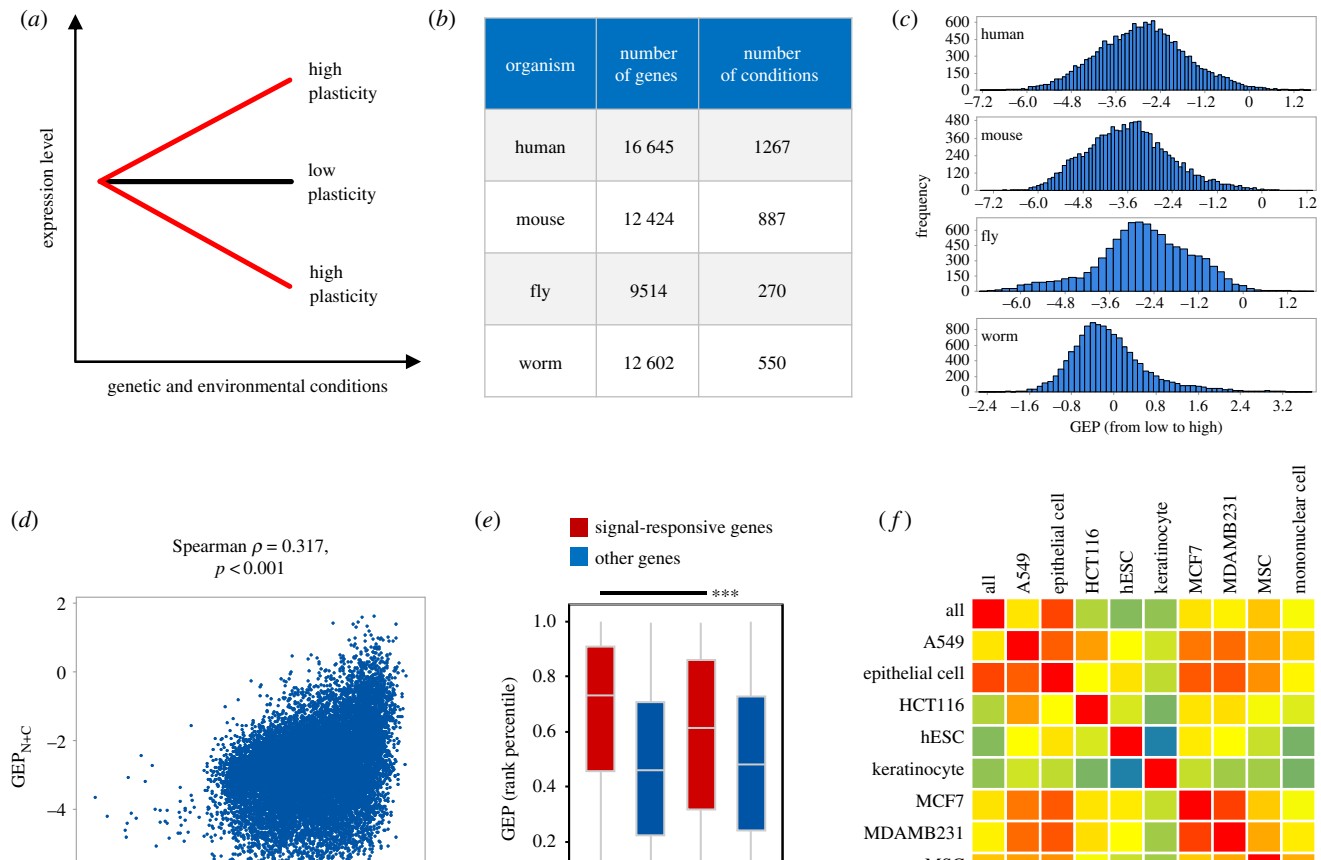

**Figure 1.** Quantification of GEP in four metazoan species. (*a*) Schematic diagram shows the definition of GEP based on changes in expression levels across genetic and environmental conditions. Genes with a high GEP exhibit dynamic expression across conditions, while genes with a low GEP exhibit stable expression. (*b*) Summary of data sources. The figure shows the species, numbers of genes and numbers of conditions used for quantifying GEP. (*c*) Distribution of GEP in four species. (*d*) Correlation between $GEP_{N+C}$ and $GEP_N$. For $GEP_N$, fold changes of gene expression were quantified between expression level in a sample and the average level across all samples. (*e*) Signal-responsive genes exhibited higher $GEP_{N+C}$ than $GEP_N$. To compare the GEP level of the same gene calculated using different expression datasets, GEP levels were normalized as the percentile rank across all genes. *** denotes Wilcoxon signed-rank test, $p < 0.001$. (*f*) Heatmap shows the correlation coefficient between global and cell-specific GEP.

## 2.2. Expression plasticity is an intrinsic gene property associated with gene function and disease susceptibility

With GEP quantified, we next determined whether GEP is an intrinsic gene property based on two criteria. First, a gene property should be evolutionarily conserved. Consistently, we found GEP to be widely conserved between species. The strongest correlation coefficient was observed between human and mouse orthologues ($\rho = 0.46$, $p < 0.001$). Weaker but statistically significant correlations were also observed between distantly related species (electronic supplementary material, figure S2*a* and table S4). Second, a gene property should measure a distinct gene feature. Gene expression level and broadness (also known as tissue specificity) are two important properties that quantify the abundancy and breadth of gene expression [16,17]. Our analysis showed GEP to be poorly correlated with expression level (electronic supplementary material, figure S2*b* and table S5; $\rho = 0.079$ and 0.179, respectively, for two different datasets) and expression broadness (electronic supplementary material, figure S2*c* and table S5; $\rho = 0.077$ and 0.080, respectively, for two different measurements of

expression broadness), suggesting GEP indicates an orthogonal feature of gene expression. Together, the above results confirm that expression plasticity is an intrinsic gene property indicating the changeability of gene expression.

We next sought to determine the physiological implications of GEP. First, we established that GEP is significantly associated with specific gene functions. Consistent with intuition, analysis of gene functional annotations [18] revealed that genes with high GEP were significantly overrepresented in biological processes that are important for cellular flexibility, such as inflammatory response, immune response and response to drugs (figure 2*a*). In addition to global functional classifications, we compiled lists of genes with well-defined physiological functions and confirmed again that GEP and biological function were nicely consistent (electronic supplementary material, table S6). Specifically, homeobox genes, which are critical for specifying cell fate and body plan, exhibited significantly lower GEP (figure 2*b*; electronic supplementary material, table S6). So did hormones and their receptors, whose function is crucial for growth and development in a dosage-sensitive manner (figure 2*c*; electronic supplementary material, table S6). Conversely, the GEP of innate immune genes was significantly higher than that of other genes (figure 2*d*; electronic supplementary

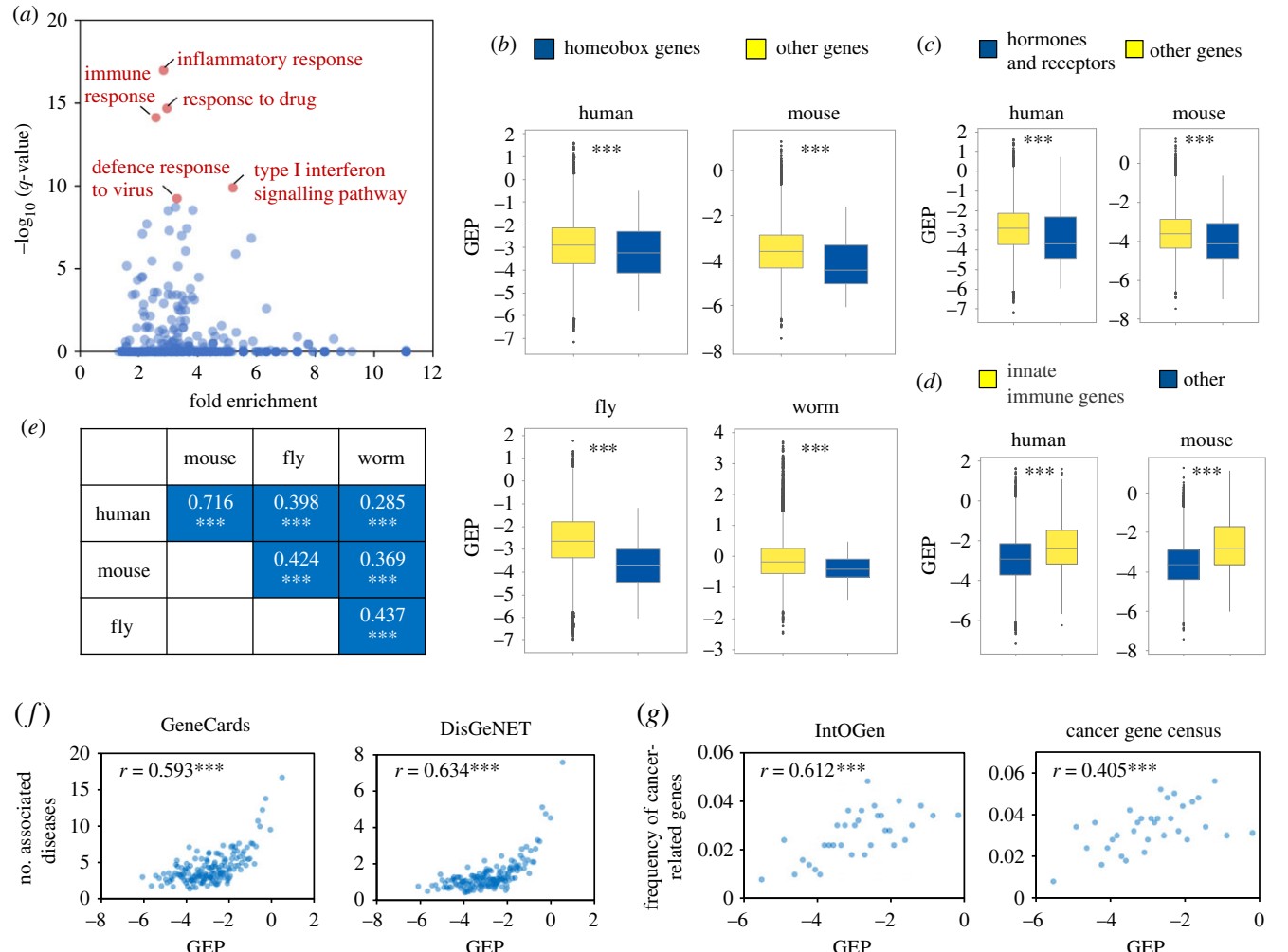

**Figure 2.** Expression plasticity is associated with gene function and disease susceptibility. (*a*) Enrichment biological processes for genes with high GEP. Scatter plot shows the fold enrichment and *q*-values of all biological processes for genes exhibiting high GEP (top 10%, *n* = 1643). The top five most significantly enriched processes are marked in red. (*b–d*) Homeobox genes (*b*) and hormones and receptor genes (*c*) exhibited significantly lower GEP; innate immune genes exhibited significantly higher GEP (*d*). *** denotes Mann–Whitney *U* test, *p* < 0.001. (*e*) Ontology term GEP is highly correlated between species. GEP values for each ontology term were calculated by averaging the GEP of all genes belonging to that term. The figure shows the correlation coefficients between ontology term GEP values for all cross-species comparisons. *** denotes *p* < 0.001. (*f*) Genes with high GEP are implicated in disease susceptibility. Genes were ranked by GEP and the average number of associated diseases calculated for each bin (100 genes). Gene–disease associations were extracted from GeneCards [19] and DisGeNet [20]. (*g*) Genes with high GEP are implicated in cancer. Genes were ranked by GEP and the frequency of cancer-related genes was calculated for each bin (500 genes). Cancer-related genes were obtained from IntOGen [21] and the Cancer Gene Consensus [22].

material, table S6), consistent with their important roles in immune response. Importantly, the association of gene function with characteristic GEP level was highly conserved between species; the average GEP values of genes in given Gene Ontology (GO) terms (GEP_GO) were significantly correlated for all pairwise species comparisons (figure 2*e*; electronic supplementary material, table S7), suggesting that genes with a similar function tend to have a concordant GEP level across species.

Second, we revealed that genes with high GEP are implicated in disease susceptibility. Considering that genes with high GEP would confer flexibility, we tested whether their malfunction would be more frequently implicated in disease. We found that the GEP values of human genes were positively correlated with their propensity for disease association. The average number of diseases associated with a gene increased with expression plasticity (figure 2*f*; electronic supplementary material, table S8). Similarly, the chances of a gene being cancer-related positively correlated with GEP (figure 2*g*; electronic supplementary material, table S8); the frequency of cancer-related genes increased from 1–2% in genes with low plasticity to 4–6% in those with high plasticity.

Altogether, we have systematically quantified GEP in four metazoan species and demonstrated it to be an intrinsic gene property with important biological implications. Low-plasticity genes tend to function in cellular processes demanding high stability, whereas high-plasticity genes are enriched in cellular processes demanding high flexibility. In particular, genes with high expression plasticity tend to be crucial for maintaining organismal fitness, especially under challenges such as disease conditions. The broad conservation and important physiological implications of expression plasticity indicate its magnitude is a specific trait under tight regulation. In the following sections, we investigate the genomic regulation of expression plasticity.

## 2.3. Influence of core promoter *cis*-elements on gene expression plasticity

We began by investigating the influence of promoter elements on GEP. It has been reported that yeast genes with a TATA box motif exhibit significantly higher GEP than those without it [3,23]. This trend was nicely recapitulated in our data. As

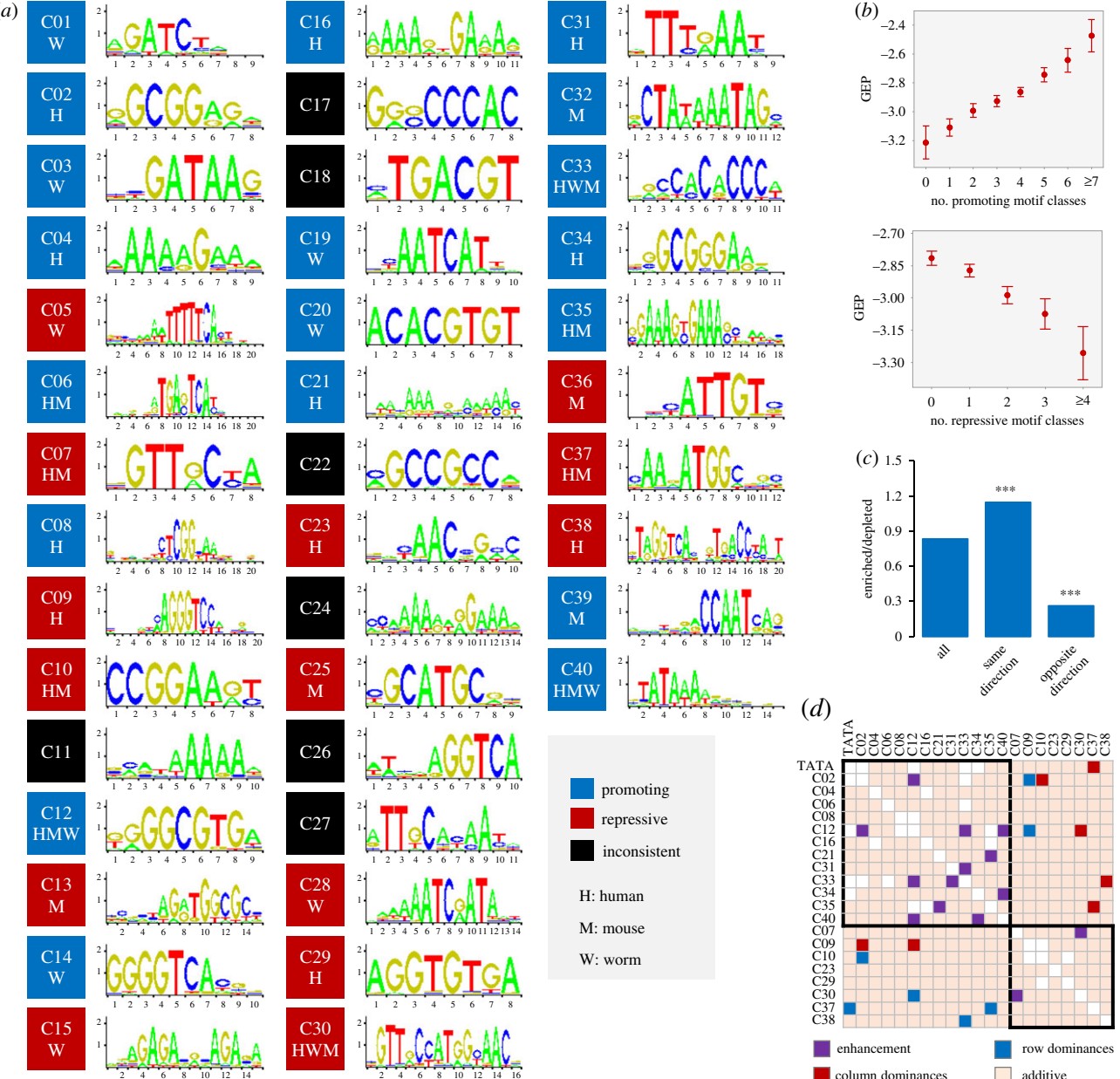

**Figure 3.** Influence of promoter elements on GEP. (a) The figure shows the sequence logo and effects of motif classes associated with GEP. Black indicates motif classes with inconsistent effects; blue and red indicate the 33 motif classes that have GEP-promoting or -repressive roles in corresponding species, respectively. Because individual motifs within motif classes exhibit high sequence similarity, representative motif sequence logos are used. (b) Cumulative effects of motif classes on human GEP. The figure summarizes the changes in GEP as more GEP-promoting (top) and -repressing (bottom) motif classes occur in gene promoters. See electronic supplementary material, figure S3c for results from mouse and worm. (c) Optimization of motif co-occurrence in human promoters. Co-occurrence was evaluated for all motif pairs that affected GEP in the same or opposite directions. A motif pair was considered enriched or depleted if their observed co-occurrence was significantly higher or lower than expected (Fisher's exact test, $p < 0.05$), respectively. *** denotes a ratio significantly different from that of all motif pairs (Fisher's exact test, $p < 0.001$). See electronic supplementary material, figure S3d for results from mouse and worm. (d) Combinatorial effects of DNA motifs on human GEP. Matrix shows additive (light orange), enhancement (purple) and dominance (blue and red) relationships between all pairwise motif classes. Black boxes highlight motif classes that affect GEP in the same direction. See electronic supplementary material, figure S3f for results from mouse and worm.

shown in electronic supplementary material, figure S3a, the frequency of TATA box-containing promoters increased in genes with high GEP.

We then examined whether there exist additional promoter elements influencing plasticity. To facilitate cross-species comparison, we used a large collection of well-characterized DNA motifs sourced from multiple species [24]. This approach identified 141 GEP-associated motifs (electronic supplementary material, table S9), including a TATA box-like motif (Mann–Whitney U-test, Benjamini–Hochberg corrected $p < 0.01$). An observed lack of GEP-associated motifs in *Drosophila* might have been due to underrepresentation of that species in the motif dataset. Because many motifs exhibited high sequence similarity, we further collapsed similar motifs into a single motif class. This yielded 40 unique classes, 33 of which exhibited consistent effects on GEP (electronic supplementary material, table S9). Respectively, 19 and 14 types of motifs promoted or repressed GEP, and nine classes showed conserved roles in two or more species (figure 3a). As the DNA elements in promoters might not be independent in their presence and function, we also determined the contribution of each of the

33 GEP-associated motif classes after controlling for the influence of all other classes. To do so, we performed stepwise model selections by calculating the Akaike information criterion (AIC). The stepwise AIC procedure iteratively adds and removes model predictors (motif classes) in order to identify the subset with the best performance (lowest AIC score). Only one motif class (C35) was removed from the model, indicating that most motif classes influence GEP after controlling for the effects of others (electronic supplementary material, figure S3b). Consistently, the effects of motif classes that influenced GEP in the same direction were accumulative; GEP levels changed progressively as the number of same-direction motif classes increased (figure 3b; electronic supplementary material, figure S3c). These results suggest that many promoter cis-elements in addition to the TATA box influence GEP non-redundantly.

One promoter may contain both GEP-promoting and -repressing elements; therefore, is the presence of motif classes with opposite effects in the core promoter optimized to coordinate their influence on GEP? If so, we predict that the co-occurrence of motif classes that influence GEP in the same direction would be enriched (higher than expected), whereas the co-occurrence of those working in opposite directions would be depleted (lower than expected). Indeed, for all pairwise motif classes, the ratio of enriched to depleted was significantly higher for those influencing GEP in the same direction than those with opposite directions of effect (figure 3c; electronic supplementary material, figure S3d). Hence, cis-element promoter architecture is optimized to maximize its effect on GEP.

To further characterize the interactions between DNA elements, we analysed the combinatorial effects between pairwise motif classes to identify three types of interactions, namely additivity, enhancement and dominance (electronic supplementary material, figure S3e). Most pairwise motif classes (88%, 87% and 91% in human, mouse and worm, respectively) function additively to influence GEP. The observed GEP values of promoters containing both motif classes were similar to those expected from summing the effects of individual motif classes (figure 3d; electronic supplementary material, figure S3f). Interestingly, a small number of motifs exhibited non-additive interactions. In humans (figure 3d), we identified seven pairs of motif classes that demonstrated enhancement interactions in which the observed effects in promoters with both motif classes were significantly stronger than expected. In addition, seven pairs of motif classes exhibited dominance interactions in which the effect of one motif class was masked by that of the other.

The above analysis of the contributions of promoter cis-elements to expression plasticity reveals that many promoter cis-elements function individually or synthetically to influence expression plasticity. A DNA element often regulates gene expression through the binding of trans-acting proteins, such as transcription factors. However, our knowledge of the association between any given cis-element and its corresponding regulatory proteins is fairly limited. This uncertainty prevented us from performing a systematic analysis of how motifs regulate expression plasticity via regulatory proteins. To circumvent this complexity, we next adopted a regulatory protein-centric strategy in order to investigate whether certain trans-acting proteins regulate expression plasticity.

## 2.4. Promoter binding of specific trans-acting proteins is associated with gene expression plasticity

The availability of genome-wide binding patterns for many regulatory proteins provides rich opportunities to elucidate their functions in GEP. In particular, the Encyclopedia of DNA Elements (ENCODE) [25] has mapped in vivo binding regions for a large collection of human regulatory proteins in many samples, allowing us to systematically evaluate the potential functions of trans-acting proteins. If a regulatory protein influences GEP, its target genes would exhibit significantly higher or lower values than those it does not bind. Notably, where GEP describes a constant feature of gene expression, the occupancy of regulatory proteins at target genes is highly context-specific. We, therefore, required that the protein–GEP association be consistently detected in a majority of samples.

Through screening genome-wide binding data for 159 regulatory proteins in 505 samples (electronic supplementary material, table S10), we identified six (CEBPB, CTCF, RAD21, RELA, TBP and TCF7L2) and four (NR2C2, GABPA, SIX5 and ZNF143) proteins that were positively and negatively associated with GEP, respectively (figure 4). Notably, the list of GEP-promoting regulators includes TBP, the TATA box binding protein, consistent with the observation that TATA box-containing promoters exhibit high GEP [3]. We also found that co-binding regulatory proteins exhibited consistent effects on GEP. For example, CTCF, the CCCTC-binding factor, and its interacting protein RAD21, the Scc1 component of the cohesin complex [27], were both associated with higher GEP (figure 4a). Significantly, some of the identified proteins had effects consistent with the motif analysis results. For example, the effects of CEBPB and GABPA were captured by both motif-centric and regulatory protein-centric analyses (electronic supplementary material, table S9 and figure 4). Similarly, the motif MA0088.2 related to ZNF143 was associated with a lower GEP (adjusted $p = 0.011$), which, while not passing the cut-off for significance (adjusted $p < 0.01$), was consistent with the results of protein occupancy analysis.

## 2.5. Validating the role of trans-acting proteins

To test whether the above results were simple correlations or reflected causal regulatory relationships, we used gene expression datasets from knockout experiments to validate the causal roles of transcription factors. After careful curation, we identified five expression datasets for CTCF, RAD21, RELA ($n = 2$) and TCF7L2 [28–32] that met the following criteria: first, the regulatory protein was mutated (null mutants) or knocked out; and second, genome-wide gene expression was assayed for at least two different conditions for both control (CTR) and loss of function (LOF) mutants.

Because all four proteins were predicted to be GEP-promoting (figure 4a), we expected that their inactivation would induce a significant reduction in GEP. For each dataset (figure 5a), we compared the magnitude of expression changes for very dynamically expressed genes (top 20% of genes with highest expression changes) as an approximation of GEP for a condition. We found a significant reduction (Mann–Whitney U-test, $p < 0.001$ for all cases) in the magnitude of expression changes for all four regulatory proteins (four out of five

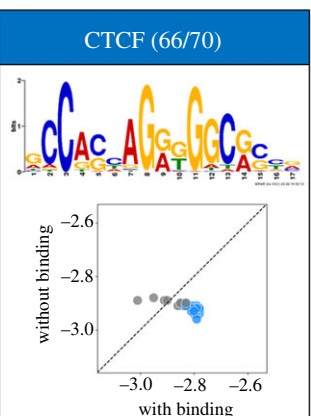
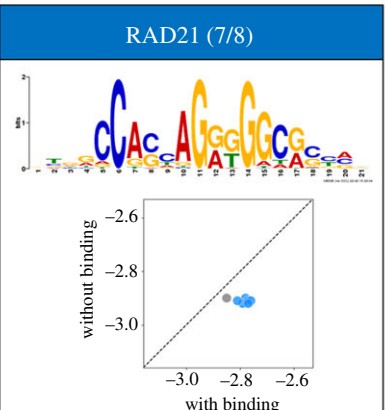
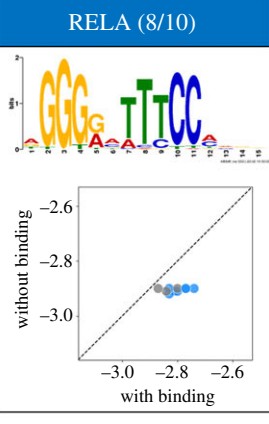
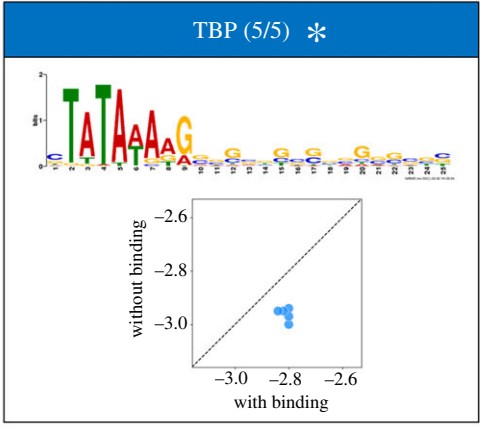
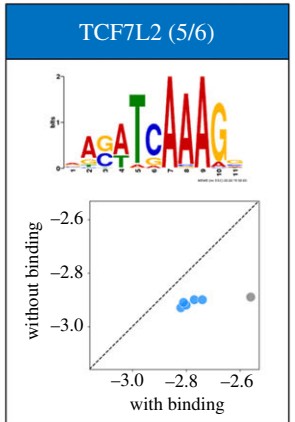
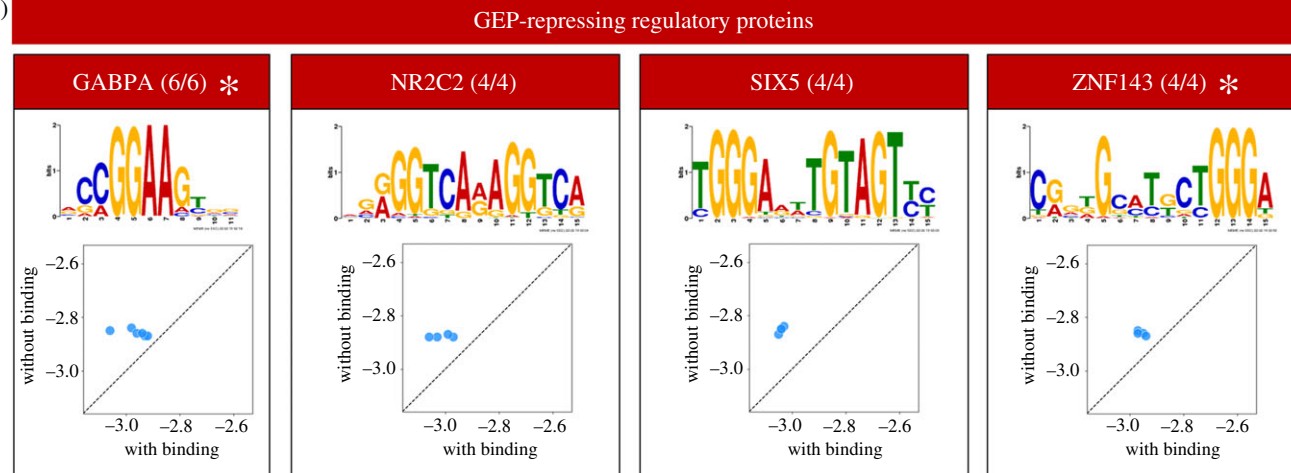

**Figure 4.** Influence of regulatory proteins on GEP. GEP-promoting (*a*) and -repressing proteins (*b*) were defined as those whose target genes exhibited significantly (Mann–Whitney *U*-test, Benjamini–Hochberg corrected $p < 0.01$) higher or lower GEP than other genes. The figure shows the regulatory protein name, the number of samples with significant GEP association, the total samples examined and the preferential binding motif as defined by Factorbook [26]. Star indicates that the protein-binding result is consistent with that from motif analysis. Scatterplot compares GEP between genes with (*X*-axis) and without (*Y*-axis) binding in the promoter region (from −500 to +500 relative to the TSS) of regulatory proteins having a promoting (*a*) or repressive (*b*) role. Blue indicates $p < 0.01$ (Mann–Whitney *U*-test, Benjamini–Hochberg corrected).

datasets) (figure 5*b–e*). Of the two datasets available for *RELA*, one was not consistent with our prediction (data not shown).

Furthermore, we compared the magnitudes of expression change among genes that were dynamically expressed in both genotypes and showed the same directions of expression change (co-upregulation or co-downregulation). Again, we found that genes generally exhibited lower magnitude expression changes following the inactivation of regulatory proteins (Wilcoxon signed-rank test, $p < 0.001$ for all cases). The ratio of genes showing lower magnitude to those showing

higher magnitude ranged from 1.4 to 3, which was significantly greater than expected (figure 5*b–e*, Fisher's exact test, $p < 0.001$ for all cases). It is worth noting that, while the predicted roles of these proteins were made using human data, the perturbation experiments performed using other organisms nicely validated their roles. Together, the above results reveal a previously unrecognized function for certain sequence-specific regulatory proteins in controlling expression plasticity.

Of these regulatory proteins, CTCF and RAD21 are of particular interest. Recent findings have revealed that CTCF and

royalsocietypublishing.org/journal/rsob    Open Biol. 9: 190150

(a)

| regulatory proteins | organism | GEO accession | perturbation | conditions |
|---|---|---|---|---|
| CTCF | mouse | GSE38673 | Nex-Cre-mediated conditional knockout | cerebral cortex at P7 and hippocampus at P7 |
| RAD21 | zebrafish | GSE18795 | null mutant $rad21^{nz171}$ | 24 hpf and 48 hpf |
| RELA | mouse | GSE79233 | tissue-specific *RELA* knockout | with and without TNF-α treatment |
| RELA | mouse | GSE11963 | $RELA^{-/-}$ mouse embryonic 3T3 fibroblasts | LTβR stimulation, 0 h and 10 h |
| TCF7L2/TCF4 | mouse | GSE41284 | disrupted DNA-binding HMG box, null mutant | liver sample, fasted and fed |

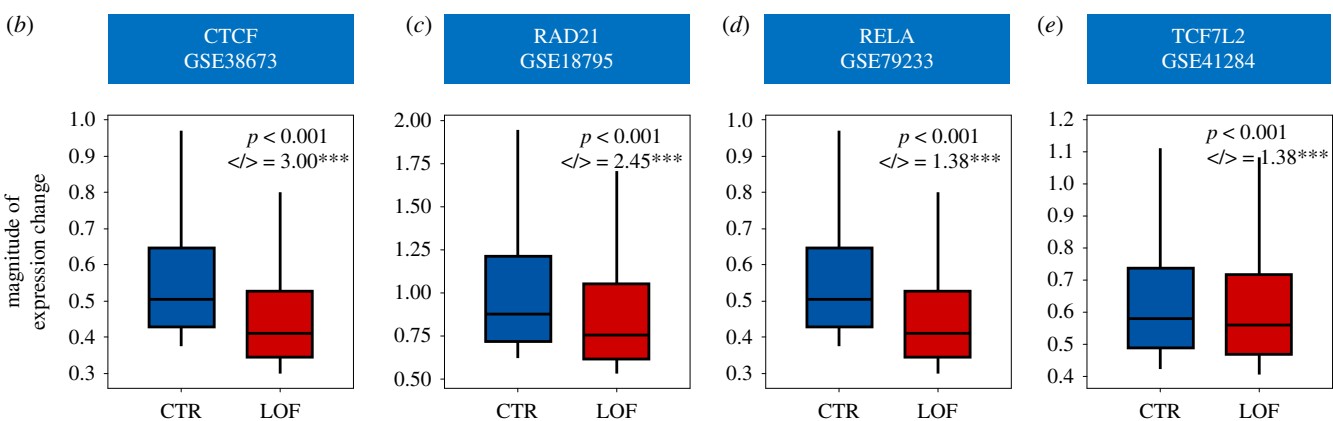

**Figure 5.** Perturbing regulatory proteins affects the magnitude of gene expression changes. (*a*) Gene expression datasets used to validate the influence of regulatory proteins on GEP. (*b–e*) Loss of function of four regulatory proteins predicted to promote expression plasticity reduced the magnitude of gene expression changes. Absolute log2-fold changes of gene expression between two conditions were used to approximate changes in GEP. Box plots show magnitudes for the mostly dynamically expressed genes (top 20%) in control (CTR) and loss of function (LOF) mutants. Gene numbers for each dataset are: $n = 1817$ for CTCF (*b*), $n = 1130$ for RAD21 (*c*), $n = 675$ for RELA (*d*) and $n = 2114$ for TCF7L2 (*e*). </> indicates the ratio of genes showing lower change magnitude to those showing higher magnitude. Only genes that exhibited expression changes in the same direction between two conditions (co-upregulation or co-downregulation) were considered. *** denotes a ratio significantly different from expected (Fisher's exact test, $p < 0.001$).

the associated cohesin complex could mediate higher order chromosome folding and chromatin interactions [33]. The binding of CTCF and cohesin at genomic sites defines the physical contact points (anchor) for the formation of a chromatin loop structure that is important in transcriptional regulation [34]. It was reported that the CTCF/cohesin-bound anchor regions exhibit significantly higher transcriptional activity than those located in the loop region [35]. As related above, both genome-wide prediction and perturbation data support a positive role for CTCF and RAD21 in regulating expression plasticity (figures 4*a* and 5*b,c*). Another subunit of cohesion, SMC3, did not pass our stringent cut-off to identify regulators of GEP, but was also associated with higher expression plasticity in three out of four examined samples (electronic supplementary material, table S10). These results inspired us to test whether CTCF and the cohesin complex regulate GEP through a chromatin topology-based mechanism (electronic supplementary material, figure S4*a*). If it does, we could make two predictions: first, genes whose promoters are located in the anchor regions of a chromatin loop would exhibit significantly higher GEP than those located in loop regions; and second, CTCF binding sites outside the loop structure would not be associated with GEP. We tested this possibility using the GM12878 B-lymphocyte cell line, in which genome-wide binding data for CTCF, RAD21 and SMC3 and chromatin topology data are both available

[25,35]. Our results excluded the possibility of topology-based GEP regulation through the following observations. First, genes exhibited similar GEP values regardless of whether their promoters were located in anchor or loop regions (Wilcoxon signed-rank test, $p > 0.01$) (electronic supplementary material, figure S4*b*). Second, binding of CTCF outside the loop structure remained significantly associated with higher GEP (electronic supplementary material, figure S4*c*, Mann–Whitney $U$-test, $p < 0.001$). These findings suggest that a different mechanism accounts for the GEP-promoting role of CTCF. Indeed, CTCF is known to be a multi-functional protein and is implicated in gene regulation through diverse mechanisms [36].

## 2.6. Gene body H3K36me3, H3K79me2 and H4K20me1 modifications are associated with restricted expression plasticity

In addition to DNA sequence and transcription factors, epigenetic modifications also play prominent roles in regulating gene expression. We collected diverse epigenomic datasets and examined their relationships with GEP. For the identification of GEP-associated epigenetic signatures, a similar strategy was adopted as for GEP-associated regulatory proteins. We identified three histone modifications, mainly occurring in gene body

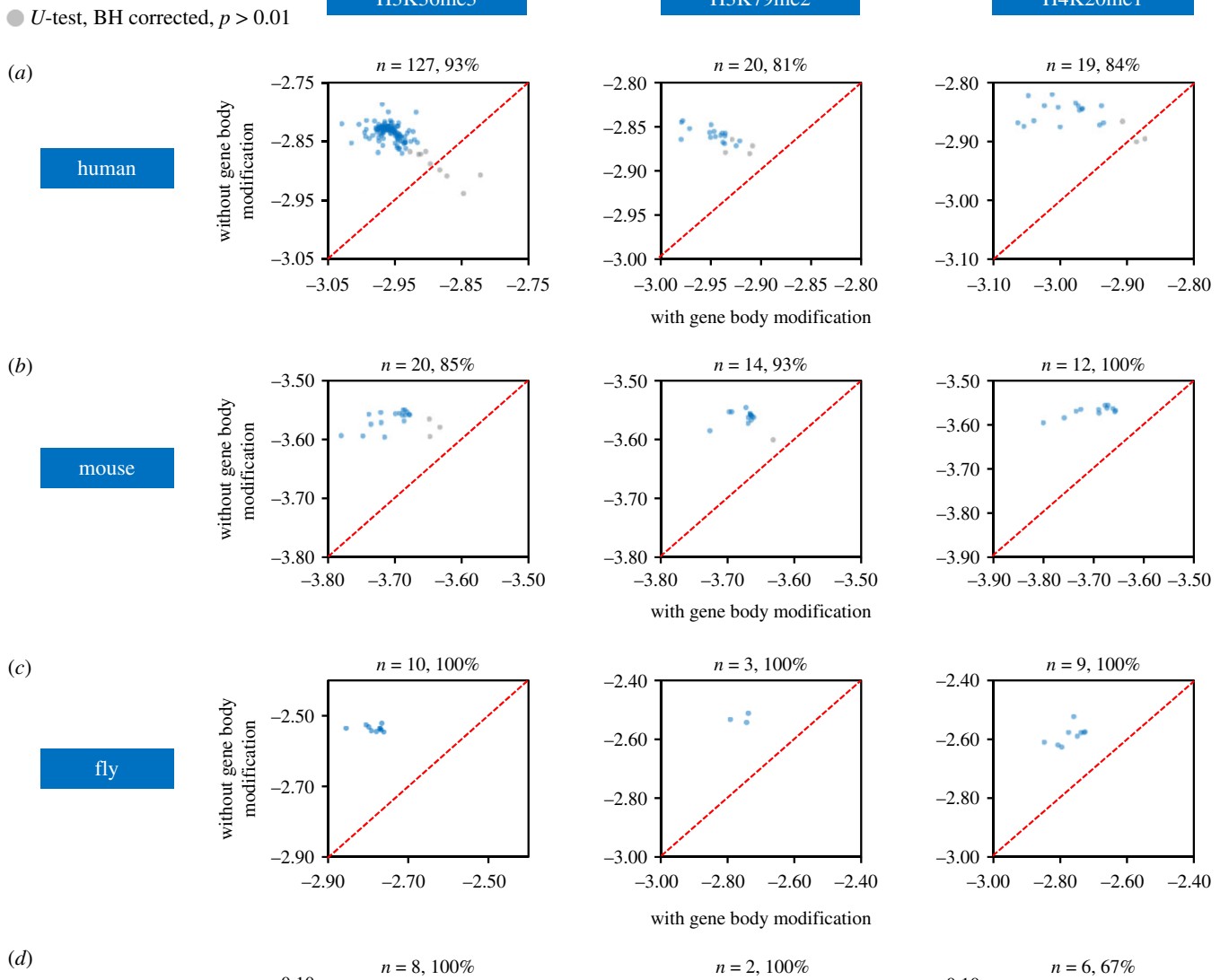

**Figure 6.** Genes enriched for gene body H3K36me3, H3K79me2 and H4K20me1 modifications exhibit restricted expression plasticity. (*a*–*d*) Gene body H3K36me3, H3K79me2 and H4K20me1 modifications restrict GEP in four species. Each scatterplot shows median GEP for genes with (*X*-axis) or without (*Y*-axis) a histone modification in the gene body region. Samples showing significant difference (Mann–Whitney *U*-test, Benjamini–Hochberg (BH) corrected $p < 0.01$) are indicated in blue. Diagonal line indicates equal GEP. The number of samples analysed and the percentage of samples with consistent differences are shown above each scatterplot.

regions (electronic supplementary material, figure S5*a* and table S11), that were strongly correlated with lower GEP: trimethylation of histone H3 at lysine 36 (H3K36me3), dimethylation of histone H3 at lysine 79 (H3K79me2) and monomethylation of histone H4 at lysine 20 (H4K20me1). Genes enriched for these modifications in the gene body exhibited significantly lower GEP than those depleted (Mann–Whitney *U*-test, Benjamini–Hochberg corrected $p < 0.01$) in a majority (greater than or equal to 80%) of the examined human samples (figure 6*a*; electronic supplementary material, table S12). Genes with high GEP values exhibited lower frequencies of these histone modifications in the gene body (electronic supplementary material, figure S5*b* for representative examples). Importantly,

the repressive effects of H3K36me3, H3K79me2 and H4K20me1 on GEP were consistently detected in all four metazoan organisms (figure 6*b*–*d*; electronic supplementary material, table S12). We did not identify combinatorial effects of the above modifications on GEP as genes with two types of modifications did not exhibit significantly lower GEP levels than genes with only one type of modification, and genes with all three types of modification did not show significantly lower GEP levels than those with two or one type of modification (electronic supplementary material, figure S5*c* and table S13). Collectively, these results predict that gene body H3K36me3, H3K79me2 and H4K20me1 modifications function to reduce expression plasticity.

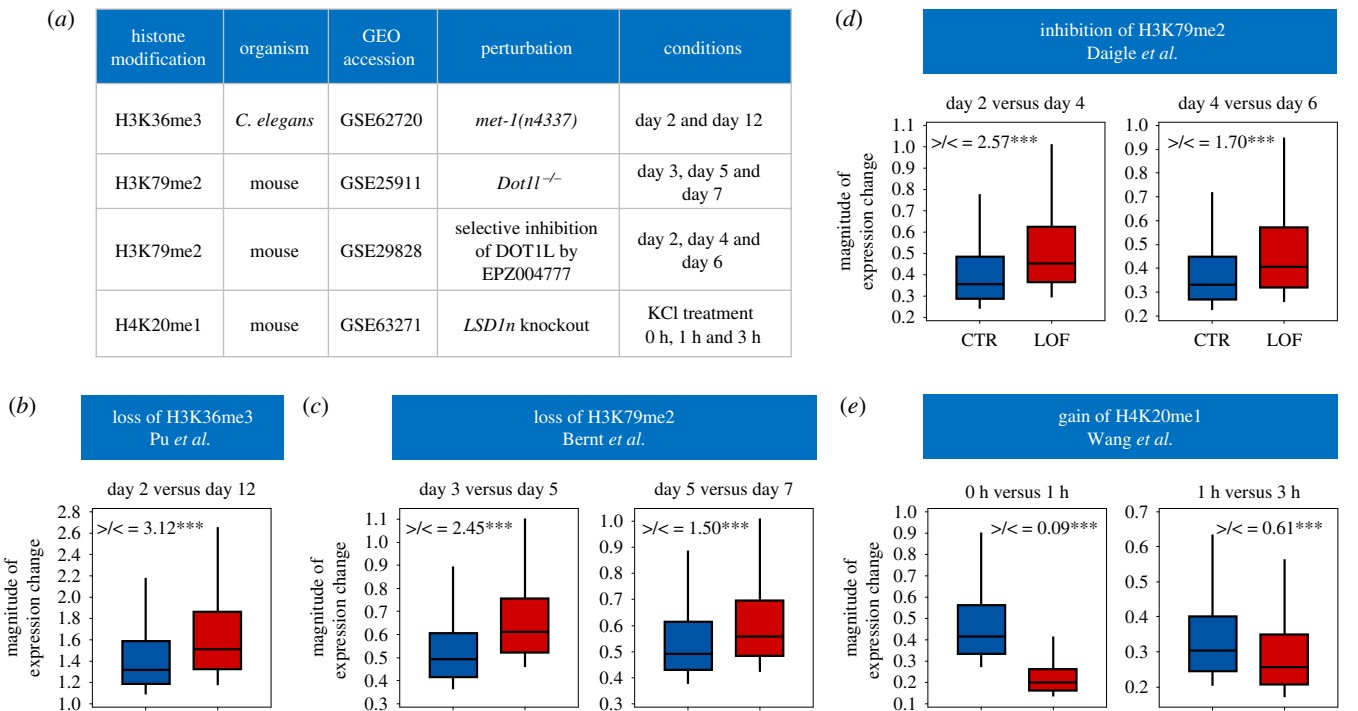

**Figure 7.** Perturbing histone modifications affects magnitude of gene expression changes. (*a*) Gene expression datasets used to validate the effect of histone modifications on GEP. (*b*–*e*) Perturbation of H3K36me3 (*b*), H3K79me2 (*c,d*) and H4K20me1 (*e*) induced expected changes in the magnitudes of gene expression changes between two conditions. Absolute log2-fold changes of gene expression between two conditions were used to proximate GEP changes. Box plots show magnitudes for the most dynamically expressed genes (top 20%) in control (CTR) and loss (LOF) or gain (GOF) of function mutations. Gene numbers for all datasets are: n = 535 from Pu *et al.* [37]; n = 141 (day 3 versus day 5) and n = 190 (day 5 versus day 7) from Bernt *et al.* [40]; n = 1483 (day 2 versus day 4) and n = 460 (day 4 versus day 6) from Daigle *et al.* [41]; and n = 1062 (0 h versus 1 h) and n = 1106 (1 h versus 3 h) from Wang *et al.* [42] >/< indicates the ratio of the number of genes exhibiting larger magnitudes of change to those exhibiting smaller magnitudes for genes having the same direction of expression change between two conditions. *** denotes a ratio significantly different from expected (Fisher's exact test, p < 0.001).

## 2.7. Validating the role of histone modifications

To determine a causal role for the above histone modifications, we analysed the change of GEP in mutants in which histone modifications levels were perturbed. We mined the literature and collected four relevant datasets (figure 7*a*) within which global gene expression was assayed for multiple conditions in both wild-type and mutants of the corresponding enzyme that adds or removes a modification of interest. We expected that the loss or gain of these histone modifications would induce a respective increase or reduction in GEP.

For H3K36me3, we used a *C. elegans* dataset in which the *met-1* gene that encodes the H3K36 methyltransferase was mutated [37]. MET-1 is required for maintaining the global H3K36me3 level [38], and in *met-1* mutants the modification level is reduced by over 90% [37]. We compared the magnitudes of global gene expression changes (day 2 versus day 12) between control samples and *met-1* mutants to approximate GEP. To specifically analyse the contribution of H3K36me3, we focused only on those genes enriched in H3K36me3 in the gene body regions of control samples at both day 2 and day 12. As predicted by our computational analysis (figure 6), the magnitude of gene expression change was significantly increased following the loss of H3K36me3 (figure 7*b*). We also compared the magnitudes of expression change for genes that showed identical directions of change (co-upregulation or co-downregulation). The ratio of genes showing higher magnitude to those showing lower magnitude was significantly higher than expected (3.12 versus 1, Fisher's exact test, p < 0.001). These findings confirm a GEP-repressive role for H3K36me3. Interestingly,

the authors of this dataset reported a similar function for H3K36me3 in restricting age-dependent gene expression changes, which impacts the *C. elegans* lifespan [37]. In addition, they identified similar results using *Drosophila* data, suggesting that this phenomenon is conserved across metazoans. H3K36me3 is known to be enriched at actively transcribed genes [39]; the increased magnitudes of expression change induced by inactivation of H3K36me3 thus cannot be simply explained by its role in transcriptional activation.

For H3K79me2, we identified two mouse datasets in which the *DOT1L* gene that encodes the histone H3K79 methyltransferase was either knocked out [40] or selectively inhibited [41]. In both mutants, the level of H3K79me2 was dramatically reduced. As above, we compared the magnitudes of expression changes in multiple conditions and found them to be significantly increased (figure 7*c,d*). In genes showing identical directions of expression change, the ratio of higher magnitude to lower magnitude changes was significantly higher than expected (figure 7*c,d*). To specifically measure the effect of H3K79me2, we focused on genes enriched for the modification in the gene body region in knockout control samples. For the inhibition dataset, genome-wide H3K79me2 modification data were not available; therefore, our analysis was applied to all genes.

For H4K20me1, we identified one mouse dataset in which a neuron-specific isoform of *LSD1* (*LSD1n*), an H4K20 demethylase, was conditionally knocked out [42]. This resulted in a significant and specific elevation of H4K20me1 levels [42]. We expected that this gain of function for H4K20me1 would induce a reduction in GEP. Consistently,

royalsocietypublishing.org/journal/rsob  Open Biol. 9: 190150

loss of *LSD1n* resulted in significantly decreased magnitudes of gene expression change (figure 7*e*). Among genes showing identical directions of expression change, the ratio of those with higher magnitude changes to lower magnitude changes was significantly lower than the expectation (figure 7*e*).

Collectively, the above data reveal that certain gene body histone modifications determine GEP, in addition to their well-established functions in regulating gene transcription [43], splicing [39,44], DNA replication [45] and DNA repair [46].

# 3. Discussion

## 3.1. Expression plasticity is a conserved gene property

Although expression plasticity has been quantified under a variety of genetic and environmental conditions in different species, there is widespread correlation of expression plasticity between orthologues (electronic supplementary material, figure S2*a*). Importantly, expression plasticity is widely associated with particular gene functions, and this association is also evolutionarily conserved (figure 2*e*). These findings suggest that the expression plasticity of different genes is optimized to exert biological functions and that plasticity is a specific target of regulation through evolutionarily conserved mechanisms. An important further question is to what extent expression plasticity varies across cell types. Owing to the relative scarcity of cell-specific expression change datasets, we performed a meta-analysis on combined data from cell lines, tissues and whole organisms. As per the preliminary results shown in figure 1*f*, different cell types seem to exhibit differential expression plasticity. However, the relatively small number of conditions included (fewer than 100 for most cases) prevents us from drawing a strong conclusion. It will be interesting to assay and compare the magnitude of gene expression changes in different cell types in response to diverse conditions. Such an assay will provide insights on the implications of expression plasticity for cellular function and on cell-specific mechanisms of plasticity regulation.

## 3.2. Independent regulation of gene expression level, plasticity and noise

Expression plasticity correlates poorly with gene expression level (electronic supplementary material, figure S2*b*), indicating that different mechanisms are used to independently regulate various properties of gene expression. As expected, we found gene expression levels and plasticity to be regulated by distinct genetic and epigenetic mechanisms. As for expression plasticity, we identified DNA motifs that are associated with expression levels in multiple cell lines. However, none of the motifs that influenced expression plasticity were associated with expression level (data not shown). In addition, a genome-wide analysis of the contribution of 38 types of histone modifications to gene expression levels in human cells revealed that H2BK5ac, H3K27ac, H3K79me1 and H4K20me1 are the most important histone modifications for predicting expression levels, all of which positively correlate with expression [47]. Again, none of these modifications play similar roles in regulating expression plasticity. It appears that cells use distinct regulatory programmes to separately modulate gene expression levels and plasticity.

Genome-wide studies in yeast have revealed GEP and expression noise to be highly correlated [1,3,6,23,48]. Genes with high expression variability among isogenic cells (high expression noise) tend to exhibit high magnitudes of expression change in response to stimuli (high expression plasticity). This thus suggests the existence of a common mechanism underlying expression noise and plasticity. Consistent with this, the TATA box promoter element is positively correlated with both GEP and cell-to-cell expression noise [4,49]. Recent studies have also identified certain types of histone modification as significantly associated with expression noise [49,50], some of which were independently identified here to regulate expression plasticity. However, closer examination of the relationship between noise and plasticity demonstrates that the coupling between the two is highly conditional and evolvable in both yeast [4] and *Escherichia coli* [51]. For example, the correlation between noise and plasticity is disfavoured for essential genes and haploinsufficient genes, whereby certain genes could have both high plasticity and low noise. It is unclear whether the noise–plasticity coupling also exists in higher organisms. We evaluated the noise–plasticity relationship using a recently published single-cell transcriptome dataset from mouse embryonic stem cells under three culture conditions [52]. Where expression noise had a relatively high correlation coefficient between conditions (electronic supplementary material, figure S6*a*), the correlation between noise and plasticity was significantly lower in all conditions (electronic supplementary material, figure S6*b*). This suggests the coupling between noise and plasticity is not a general rule in higher organisms. Furthermore, while H3K27me3, H3K4me1 and H3K9ac are associated with high expression noise in mouse embryonic stem cells [49], we found that these modifications do not affect expression plasticity. Interestingly, H3K36me3, H3K79me2 and H4K20me1 have been independently identified as significantly associated with restricted expression noise [50] and plasticity (figure 6). However, the expression noise data [53] and plasticity data used in each study are not correlated (electronic supplementary material, figure S6*c*; $\rho = 0.023$, $p = 0.059$). This suggests that these histone modifications regulate expression noise and plasticity through distinct mechanisms.

## 3.3. Effect of histone modifications on gene expression plasticity

While many histone modifications regulate gene expression level, fewer regulate expression plasticity. Out of all 30 types of histone modifications examined, three modifications (H3K36me3, H3K79me2 and H4K20me1) were found to restrict plasticity, and none promoted plasticity. Previous works have shown that H3K36me3 functions to maintain gene expression stability and fidelity [54,55], and to maintain epigenetic memory of gene transcription in germ cells [56–58]. The negative role of H3K36me3 in regulating expression plasticity is consistent with the reported roles. Whether H3K79me2 and H4K20me1 play a similar function in the above processes is worth testing in future studies. All identified histone modifications are plasticity-restricting, raising the possibility that histone modifications may mainly function to downregulate plasticity to ensure stable and robust expression. All plasticity-restricting modifications were found to be enriched in gene body regions (electronic supplementary material, figure

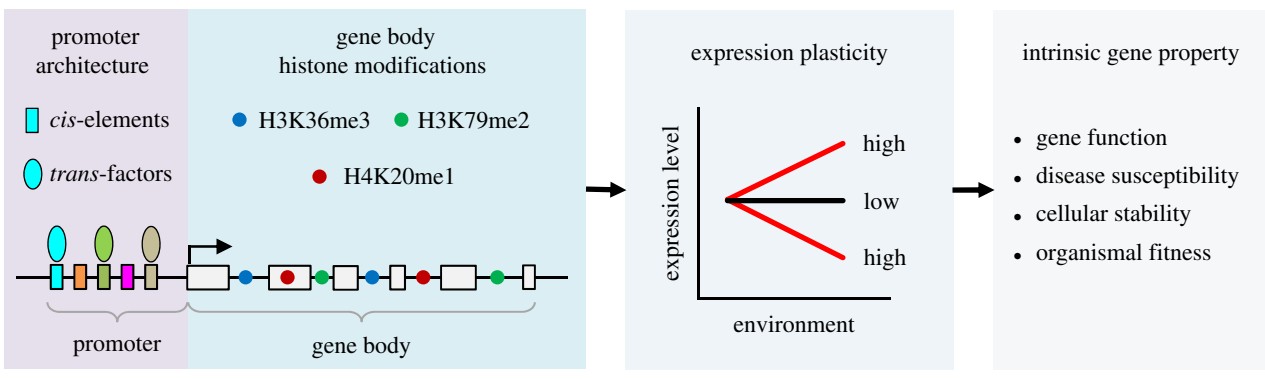

**Figure 8.** Summary of findings. Gene structure is illustrated by linked grey boxes with an arrow indicating the TSS. Rectangles indicate *cis*-elements and coloured ovals depict regulatory proteins binding in the promoter region. Circles indicate gene body histone modifications.

S5*a*), suggesting transcriptional elongation might be a regulatory target for controlling expression plasticity. A previous study showed that the *C. elegans* ZFP-1/DOT-1.1 complex, an H3K79 methyltransferase, promotes Pol II pausing [59]. In addition, inactivation of mouse *LSD1n*, a methylase that removes histone H4K20 methylation, increases Pol II pausing [42]. These findings raise the possibility that these gene body histone modifications restrict expression plasticity by inducing Pol II pausing, an important mechanism for regulating gene expression [60]. The Global Run-on Sequencing method (GRO-seq), which maps the binding sites of transcriptionally active Pol II, has been widely used to accurately quantify Pol II elongation and pause-release [61]. Using human GRO-seq data [62–64], we found that genes with or without Pol II pausing exhibited similar expression plasticity (electronic supplementary material, figure S7 and table S12). Pol II pausing index values [65,66], a parameter for quantifying Pol II pausing, were indistinguishable for genes with high and low GEP. Whether gene body histone modifications function through affecting other aspects of transcriptional elongation would be an important question to test in the future.

## 4. Conclusion

In summary, in this study, we systematically quantified GEP in four metazoan species and performed a comprehensive functional analysis of its properties, implications and multi-variable regulation with two major findings (figure 8). First, we revealed that expression plasticity is a conserved gene property related to gene function and implicated in disease susceptibility and cellular stability. This finding suggests that the changeability of gene expression is an intrinsic gene property with broad biological implications. Second, we identified genomic and epigenomic signatures that determine expression plasticity genome-wide. These findings significantly expand the functional repertoire of *cis*-elements, transcription factors and histone modifications in gene regulation. Together, our work provides insights into the genomic regulation of gene expression flexibility in multicellular organisms.

## 5. Methods

### 5.1. Quantification of gene expression plasticity

The method for GEP quantification used here is based on a previous study with minor modifications [3] which

represented GEP as the magnitudes of gene expression change under diverse conditions. We collected expression datasets in which genome-wide gene expression was assayed under at least two different conditions. The magnitude of gene expression change for a condition was determined as a fold change, and the average fold change across a large number of genetic or environmental conditions was used to quantify GEP. The Expression Atlas database [12] collects a large number of manually curated and uniformly processed gene expression datasets spanning many species and biological samples. In particular, its differential experiments section provides processed fold changes in gene expression across many conditions. Expression fold change datasets for human, mouse and fly genes were directly extracted from the Expression Atlas. Because of the paucity of *C. elegans* expression datasets in the Expression Atlas, worm data were obtained from a previous study in which fold changes of genes were determined across more than 400 conditions [13]. This resource has been widely used in the field to study gene co-expression regulatory networks. A considerable number of *C. elegans* genes ($n = 2395$ according to WormBase) are organized in operons, and genes within an operon are co-expressed as a polycistron from the same promoter. We excluded such genes from analysis as comparisons of plasticity between them would not be meaningful.

We calculated GEP (electronic supplementary material, table S1) using a similar method as done previously for yeast data. First, the square of log2-fold change of mRNA abundance under a given condition as compared with the control was calculated to represent expression change under a condition. Second, expression changes were averaged across all conditions. Third, because the values showed a skewed distribution, we further log2-transformed them to represent GEP. We required expression datasets to have been assayed using the same platform when a sufficient number of conditions (more than 100) were available for that platform. Otherwise, expression datasets from multiple platforms were combined to achieve the necessary condition number.

### 5.2. Benchmark genes

We collected lists of genes with well-defined biological functions as benchmark genes to determine whether the measured GEP is biologically relevant. Human signal-responsive genes ($n = 2221$; electronic supplementary material, table S2) were obtained from NetPath [14]. Fly stress-responsive genes ($n = 891$; electronic supplementary material, table S2) were defined by Girardot *et al.*

[10]. Worm genes that are critical for stress/environmental response ($n = 505$; electronic supplementary material, table S2) were curated based on phenotype data from mutants or RNA interference (RNAi)-mediated gene knockdown. Gene-associated phenotypes were retrieved from WormBase [15] using simplemine (https://wormbase.org/tools/mine/simplemine.cgi). From all phenotypic terms, we manually curated a collection of 69 phenotypes that were associated with stress/environmental response. A gene was considered required for stress/environmental response if any of those 69 phenotypes were detected in its mutants or RNAi experiments.

## 5.3. Gene orthologues

Lists of orthologous genes were downloaded from the Ensembl genome browser (https://www.ensembl.org) [67]. We only considered one-to-one orthologues between species pairs (electronic supplementary material, table S4).

## 5.4. Gene expression level and broadness

Gene expression level is defined as the average gene expression level across a large number of samples. We used expression datasets from two previous studies to measure expression level: first, the Lukk dataset in which human gene expression level was measured based on 5372 curated human microarray datasets [17], and, second, the Functional Annotation of the Mammalian Genome 5 (FANTOM5) project in which genome-wide gene expression was quantified across a large number of tissues and cell types [16]. For the FANTOM5 dataset, log2-transformed TPM (transcripts per kilobase million reads) expression data in 230 normal human tissue or cell lines were used. The most abundant transcript was used to represent the gene if multiple transcripts were associated with the same gene. Broadness of gene expression was calculated by two different methods using the FANTOM5 dataset. In the first method, broadness was quantified as the frequency of samples in which a gene is expressed. A gene was defined as being expressed in a sample if log (TPM+1) > 0.1. In the second method, broadness was quantified as the frequency of samples in which a gene is expressed specifically. Specificity of gene expression in a sample was quantified as the expression level in that sample divided by the average expression level across all samples. We defined a gene to be specifically expressed in a sample if the specificity score was higher than 2.

## 5.5. Functional enrichment analysis

Enrichment analysis of biological processes was performed for genes with high and low expression plasticity using the Database for Annotation, Visualization and Integrated Discovery (DAVID) functional classification tool (https://david.ncifcrf.gov) with default parameters [18].

## 5.6. Genes with well-defined biological function

Homeobox genes ($n = 198$ for human, $n = 152$ for mouse, $n = 77$ for fly and $n = 62$ for worm; electronic supplementary material, table S6) were defined by the HomeoDB2 classification [68]. Hormones and receptor genes ($n = 181$ and 107 for human and mouse, respectively; electronic supplementary material, table S6) were taken from Hmrbase [69]. Innate immune genes ($n = $

898 and 431 for human and mouse, respectively; electronic supplementary material, table S6) were taken from InnateDB [70].

## 5.7. Conservation of GEP–gene function associations

Conservation of GEP–gene function associations between species was determined using GO terms (http://geneontology.org) [71]. We first populated each GO term recursively to its parent terms. Only GO terms associated with 20–500 genes were considered in our analysis. Then, GEP values for each ontology term were calculated by averaging the GEP of all genes belonging to that term (electronic supplementary material, table S7).

Gene–disease association: disease genes were extracted from DisNet (http://www.disgenet.org) [20] and GeneCards (https://www.genecards.org) [19,72]. Cancer-related genes were defined by IntGOen (https://www.intogen.org) [21] and the Cancer Gene Consensus (http://cancer.sanger.ac.uk/census) [22].

## 5.8. DNA motif-centric analysis

TATA box-containing genes were identified using the FindM tool (http://ccg.vital-it.ch/ssa/findm.php) of the Signal search analysis server [73], which scanned for the presence of the TATA box motif in the genomic region from −99 to 0 relative to the TSS using default parameters. To identify GEP-associated motifs, we used the position weight matrix file for all characterized DNA motifs (CORE 2016 dataset, $n = 1014$) obtained from the JASPAR database (http://jaspar2016.genereg.net) [24]. We scanned for the presence of each motif in the core promoter regions of genes from four metazoan species (from −200 to +100 relative to the TSS) using the find individual motif occurrences (FIMO) tool of the MEME suite with default parameters [74]. A DNA motif was considered associated with expression plasticity if a significant difference (Mann–Whitney $U$-test, Benjamini–Hochberg corrected $p < 0.01$) in plasticity was detected between genes with and without the motif in their core promoters (electronic supplementary material, table S9). Because the position weight matrices of many DNA motifs are very similar, we further collapsed individual motifs into distinct classes based on matrix similarity. First, cluster information for each of the five taxonomic groups was extracted from JASPAR 2018 (http://jaspar.genereg.net/matrix-clusters), and GEP-associated motifs belonging to a given cluster were combined. Then, motif clusters were compared between taxonomic groups to further collapse similar motifs into one class. Through this process, the 141 individual motifs associated with expression plasticity were collapsed into 40 distinct motif classes.

Evaluation of pairwise combinations of DNA motifs revealed three types of relationships: additivity, enhancement and dominance. For this analysis, we predicted the expected GEP of promoters containing both motifs ($GEP_{exp}$) by summing the effects of the individual motifs and then comparing that prediction with the observed GEP values from promoters with both motifs ($GEP_{obs}$). A relationship was defined to be additive if $GEP_{obs}$ was indistinguishable from $GEP_{exp}$ ($GEP_{obs} = GEP_{exp}$). Otherwise, $GEP_{obs}$ was further compared with the GEP values of promoters containing only one of the two motifs ($GEP_{m1}$ or $GEP_{m2}$) to determine whether enhancement or dominance occurred using the rules listed in electronic supplementary material, figure S3e. Enhancement was defined as the combined

royalsocietypublishing.org/journal/rsob Open Biol. 9: 190150

effect of two motifs exerting significantly stronger influence on GEP than the sum of their separate effects, and occurs only between motifs that influence GEP in the same direction. A dominance interaction occurred when the effect of one motif was masked by that of another, and applied only in cases of motifs influencing GEP in opposite directions. Statistical significance of GEP values was determined using the Mann–Whitney $U$-test with a $p$-value cut-off of 0.05.

As done for GEP analysis, we also analysed the influence of DNA motifs on the gene expression level. We examined the contribution of 1014 motifs to expression levels using three baseline gene expression datasets sourced from the Expression Atlas, including GTEx (53 normal tissues) [75], the Illumina Body Map (16 normal tissue types) [76] and the Roadmap Epigenomics Project (57 tissue and cell lines) [77]. For each sample, we compared expression levels between genes with and without a specific motif in their core promoter regions (from −200 to +100 relative to the TSS). A motif was considered to be associated with the gene expression level if the expression levels between motif-containing and motif-less genes differed significantly (Mann–Whitney $U$-test, Benjamini–Hochberg corrected $p < 0.01$) and consistently in more than 80% of the total 126 samples analysed.

## 5.9. Regulatory protein-centric analysis

Genome-wide *in vivo* binding data for human regulatory proteins were produced by the ENCODE project [25] and extracted from the UCSC Genome Browser (https://www.genome.ucsc.edu/ENCODE) [78]. In total, these data consisted of 505 datasets containing binding data for 159 regulatory proteins in 91 cell lines (electronic supplementary material, table S10). Binding patterns were generated for a large collection of regulatory proteins in many distinct samples by ChIP-seq experiments, and uniformly processed to identify protein-binding peaks throughout the genome. To identify regulatory protein occupancy in the promoter regions of target genes, protein-bound peaks were intersected with the promoter regions of all human genes (from −500 bp to +500 bp relative to the annotated TSS defined by Ensembl Genome Browser). To determine whether the binding of a certain regulatory protein was associated with expression plasticity, we compared expression plasticity between genes with and without binding of that protein in their promoter regions. We defined a regulatory protein as being associated with expression plasticity if it met the following criteria: (i) the difference in GEP was statistically significant (Mann–Whitney $U$-test, Benjamini–Hochberg corrected $p < 0.01$), (ii) the directions of difference were consistent in at least 80% of all examined samples, and (iii) genome-wide binding was assayed in more than three distinct biological samples.

## 5.10. Histone modification analysis

Histone modification datasets were collected from the ENCODE project, Roadmap Epigenomics Project and the Gene Expression Omnibus (GEO) database. Processing for peak regions of histone modifications was performed in the original studies. To identify evolutionarily conserved histone modifications that regulate expression plasticity, we first identified candidate modifications from the human data and then examined whether they could be validated using data from other species. We used human histone modification data from the Roadmap Epigenomics Project

(http://egg2.wustl.edu/roadmap), which contained 978 datasets illustrating the genome-wide distribution of 30 types of histone modifications in 127 samples [77]. For each dataset, we downloaded the uniformly processed broad-peak and narrow-peak files to examine their enrichment in gene body regions. Only peaks with $p$-values less than $10^{-4}$ were used. If a peak overlapped with the gene body, the gene was considered to contain the modification. A histone modification was considered to regulate expression plasticity in the human genome if: (i) expression plasticity between genes with or without the modification was significantly different (Mann–Whitney $U$-test, Benjamini–Hochberg corrected $p < 0.01$), (ii) the effects of the histone modification on expression plasticity (promoting or repressive) were consistent in over 80% of examined samples, and (iii) data were available for more than three samples.

## 5.11. Magnitude of gene expression change in perturbation experiments

We searched exhaustively for gene expression datasets in which expression levels were assayed for at least two different conditions in both wild-type and mutant samples. These datasets were necessary to examine whether inactivation of regulatory proteins or histone modifications induces differential magnitudes of gene expression change between conditions. To exclude the possibility that a gene perturbation induced dramatic changes in the gene expression programme, which would make comparisons of magnitude meaningless, we only considered datasets where a majority of differentially expressed genes (greater than 60%) were shared (consistently upregulated or downregulated) between conditions in the wild-type and mutants background. Differential gene expression was determined by $t$-test (Benjamini–Hochberg corrected $p < 0.01$). The processed expression values for all microarray probes were downloaded from the National Center for Biotechnology Information GEO database for each series. To determine gene expression levels, we mapped the probes and associated values onto all protein-coding genes. Probes that matched to multiple genes were discarded. If multiple probes matched the same gene, their values were averaged. Average gene expression and fold changes were analysed using the *limma* algorithm [79].

## 5.12. Statistics

All statistical methods and corresponding $p$-values were described in the main text, figures or figure legends. Statistical analyses including Pearson correlation, Spearman rank correlation, Mann–Whitney $U$-test, Wilcoxon signed-rank test, $t$-test, AIC analysis and Fisher's exact test were performed using Python.

Data accessibility. The datasets supporting this article have been uploaded as part of the electronic supplementary material.
Authors' contributions. L.X. and Z.Z. performed data collection, bioinformatics and statistical analysis. Z.D. and F.H. conceived of the study, designed the study, coordinated the study and participated in data analysis. Z.D. wrote the manuscript with the input of other authors. All authors gave final approval for publication and agree to be held accountable for the work performed therein.
Competing interests. We declare we have no competing interests.
Funding. This work was supported by the Strategic Priority Research Program of the Chinese Academy of Sciences (grant no. XDB19000000 to Z.D.) and the National Natural Science Foundation of China (grant nos. 31571535, 31722035 to Z.D.).

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
