## [Reviewer comments · Open Biology]

Review History

RSOB-19-0150.R0 (Original submission)

Review form: Reviewer 1

Recommendation

Accept with minor revision (please list in comments)

Do you have any ethical concerns with this paper?

No

Comments to the Author

In this manuscript, the authors systematically quantified genome-wide gene expression plasticity in multiple metazoan species. They showed that gene expression plasticity under various conditions is a conserved and intrinsic characteristic of genes. The gene expression plasticity is determined by promoter structure, sequencing motif for factor binding, and histone modification patterns. Their systematic analysis across different species confirmed and generalized the previous finding from individual model organisms that gene expression plasticity is related to specific biological functions.

There are several points the authors need to address.

- 1) If I understand correctly, the 'fold change' used in the analysis is the absolute value of $\log_2(\text{fold change})$, regardless of the change directions. It needs to be addressed more precisely in the text where "expression changes across all conditions (range from >200 to 1000)" should be changed to "the sum of absolute \log_2 fold changes across all conditions".
- 2) The authors need to use expression changes (RPKM or FPKM) in addition to fold change to calculate GEP in each species. Since gene expression levels and broadness is not significantly correlated with GEP, this additional analysis should be with similar results as that using the fold change. If not, the authors need to explain.
- 3) For the correlation between histone modification and GFP, the authors addressed that H3K36me3, H3K79me2, and H4K20me1 all contributed to low GEP. Is there any interactive effect between these histone modification marks when one gene has more than one of these histone modifications on gene body region?
- 4) In addition to the results and descriptions in the manuscript and figures, the authors need to provide data analysis method which should be adequate for others to repeat the analysis.

Minors

- 1) In Figure S2B, a y-axis marker 'CEP' is missing
- 2) For Figure S5A, the histone methylation profiles should be plotted by using gene structure as the unit (i.e. plotted using transcriptional start site and the transcriptional end site, instead of 10kb downstream region).

Review form: Reviewer 2

Recommendation

Accept with minor revision (please list in comments)

Do you have any ethical concerns with this paper?

No

Comments to the Author

I enjoyed reading the manuscript. Quiet interesting approach. Overall, very good work. I have some points which maybe my not understanding or things to look out for.

Decision letter (RSOB-19-0150.R0)

22-Oct-2019

Dear Dr Du

We are pleased to inform you that your manuscript RSOB-19-0150 entitled "Multivariable regulation of gene expression plasticity in metazoans" has been accepted by the Editor for publication in Open Biology. The reviewer(s) have recommended publication, but also suggest some minor revisions to your manuscript. Therefore, we invite you to respond to the reviewer(s)' comments and revise your manuscript.

Please submit the revised version of your manuscript within 14 days. If you do not think you will be able to meet this date please let us know immediately and we can extend this deadline for you.

- 1) A text file of the manuscript (doc, txt, rtf or tex), including the references, tables (including captions) and figure captions. Please remove any tracked changes from the text before submission. PDF files are not an accepted format for the "Main Document".
- 2) A separate electronic file of each figure (tiff, EPS or print-quality PDF preferred). The format should be produced directly from original creation package, or original software format. Please note that PowerPoint files are not accepted.
- 3) Electronic supplementary material: this should be contained in a separate file from the main text and meet our ESM criteria (see <http://royalsocietypublishing.org/instructions-authors#question5>). All supplementary materials accompanying an accepted article will be treated as in their final form. They will be published alongside the paper on the journal website and posted on the online figshare repository. Files on figshare will be made available approximately one week before the accompanying article so that the supplementary material can be attributed a unique DOI.

Online supplementary material will also carry the title and description provided during submission, so please ensure these are accurate and informative. Note that the Royal Society will not edit or typeset supplementary material and it will be hosted as provided. Please ensure that the supplementary material includes the paper details (authors, title, journal name, article DOI). Your article DOI will be 10.1098/rsob.2016[last 4 digits of e.g. 10.1098/rsob.20160049].

- 4) A media summary: a short non-technical summary (up to 100 words) of the key findings/importance of your manuscript. Please try to write in simple English, avoid jargon, explain the importance of the topic, outline the main implications and describe why this topic is newsworthy.

Images

Data-Sharing

It is a condition of publication that data supporting your paper are made available. Data should be made available either in the electronic supplementary material or through an appropriate repository. Details of how to access data should be included in your paper. Please see <http://royalsocietypublishing.org/site/authors/policy.xhtml#question6> for more details.

Data accessibility section

Sincerely,

The Open Biology Team

<mailto:openbiology@royalsociety.org>

Reviewer(s)' Comments to Author:

Referee: 1

Comments to the Author(s)

In this manuscript, the authors systematically quantified genome-wide gene expression plasticity in multiple metazoan species. They showed that gene expression plasticity under various conditions is a conserved and intrinsic characteristic of genes. The gene expression plasticity is determined by promoter structure, sequencing motif for factor binding, and histone modification patterns. Their systematic analysis across different species confirmed and generalized the previous finding from individual model organisms that gene expression plasticity is related to specific biological functions.

There are several points the authors need to address.

- 1) If I understand correctly, the 'fold change' used in the analysis is the absolute value of $\log_2(\text{fold change})$, regardless of the change directions. It needs to be addressed more precisely in the text where "expression changes across all conditions (range from >200 to 1000)" should be changed to "the sum of absolute \log_2 fold changes across all conditions".
- 2) The authors need to use expression changes (RPKM or FPKM) in addition to fold change to calculate GEP in each species. Since gene expression levels and broadness is not significantly correlated with GEP, this additional analysis should be with similar results as that using the fold change. If not, the authors need to explain.
- 3) For the correlation between histone modification and GFP, the authors addressed that H3K36me3, H3K79me2, and H4K20me1 all contributed to low GEP. Is there any interactive effect between these histone modification marks when one gene has more than one of these histone modifications on gene body region?
- 4) In addition to the results and descriptions in the manuscript and figures, the authors need to provide data analysis method which should be adequate for others to repeat the analysis.

Minors

- 1) In Figure S2B, a y-axis marker 'CEP' is missing
- 2) For Figure S5A, the histone methylation profiles should be plotted by using gene structure as the unit (i.e. plotted using transcriptional start site and the transcriptional end site, instead of 10kb downstream region).

Referee: 2

Comments to the Author(s)

I enjoyed reading the manuscript. Quiet interesting approach. Overall, very good work.
I have some points which may be me not understanding or things to look out for.

Author's Response to Decision Letter for (RSOB-19-0150.R0)

See Appendix A.

Decision letter (RSOB-19-0150.R1)

31-Oct-2019

Dear Dr Du

We are pleased to inform you that your manuscript entitled "Multivariable regulation of gene expression plasticity in metazoans" has been accepted by the Editor for publication in Open Biology.

Article processing charge

Please note that the article processing charge is immediately payable. A separate email will be sent out shortly to confirm the charge due. The preferred payment method is by credit card; however, other payment options are available.

Sincerely,

The Open Biology Team

mailto: openbiology@royalsociety.org

Appendix A

Response to Referees

We would like to thank the reviewers for their kind assessment of our study, and for their valuable suggestions on how to improve the manuscript. We have revised the manuscript accordingly and we hope that the revised manuscript is improved. Detailed responses follow (red).

Referee: 1

Comments to the Author(s)

In this manuscript, the authors systematically quantified genome-wide gene expression plasticity in multiple metazoan species. They showed that gene expression plasticity under various conditions is a conserved and intrinsic characteristic of genes. The gene expression plasticity is determined by promoter structure, sequencing motif for factor binding, and histone modification patterns. Their systematic analysis across different species confirmed and generalized the previous finding from individual model organisms that gene expression plasticity is related to specific biological functions.

There are several points the authors need to address.

1) If I understand correctly, the 'fold change' used in the analysis is the absolute value of $\log_2(\text{fold change})$, regardless of the change directions. It needs to be addressed more precisely in the text where “expression changes across all conditions (range from >200 to 1000)” should be changed to “the sum of absolute \log_2 fold changes across all conditions”.

Response: We used square of \log_2 (fold change) as a measurement of expression change regardless of the directions. We have revised this sentence to be more specific.

Page-4 “the magnitude of gene expression change after treatment was quantified as square of \log_2 fold change, and the values across all conditions (range from 270 to 1,267 in different species, figure 1b) were averaged and log-transformed to represent GEP (figure S1a).”

2) The authors need to use expression changes (RPKM or FPKM) in addition to fold change to calculate GEP in each species. Since gene expression levels and broadness is not significantly correlated with GEP, this additional analysis should be with similar results as that using the fold change. If not, the authors need to explain.

Response: Thanks for pointing this out. The expression datasets used in this study to quantify GEP was generated using microarray platform rather than RNA-seq. The microarray datasets are very efficient to compare relative expression but are less reliable or accurate to represent the absolute expression values. Considering this caveat, we adopted the most common way to quantify relative expression change as \log_2 fold change and followed the same quantification strategy used previously to quantify GEP in yeast (Tirosch 2006). We have tried to collect RNA-seq datasets that measure gene expression level before and after treatments or perturbations but were unable to collect enough number of relevant samples. We acknowledge that the absolute difference of gene expression (based on RPKM or FPKM data) would provide important and orthogonal information on expression plasticity which is definitely an important question to address in future studies.

3) For the correlation between histone modification and GFP, the authors addressed that H3K36me3, H3K79me2, and H4K20me1 all contributed to low GEP. Is there any interactive effect between these histone modification marks when one gene has more than one of these histone modifications on gene body region?

Response: Thanks for the suggestion. Actually, we did analyze the potential combinatorial effect of the three histone modifications on GEP. However, the results showed the otherwise. We have now included this result in the revised manuscript (**figure S5c and table S13**).

Page 12-“ We did not identify combinatorial effects of above modifications on GEP as genes with two types of modifications did not exhibit significantly higher GEP levels than genes with only one type of modification, and genes with all three types of modification did not show significantly higher GEP levels than those with two or one type of modification (**figure S5c and table S13**)”

4) In addition to the results and descriptions in the manuscript and figures, the authors need to provide data analysis method which should be adequate for others to repeat the analysis.

Response: In the revised manuscript we expanded the Methods section to include more details on data analysis and added 9 supplementary tables to facilitate others to reanalyze the data. We believe the revised methods section should allow others to reproduce our results.

Minors

1) In Figure S2B, a y-axis marker ‘CEP’ is missing

Response: added, thanks.

2) For Figure S5A, the histone methylation profiles should be plotted by using gene structure as the unit (i.e. plotted using transcriptional start site and the transcriptional end site, instead of 10kb downstream region).

Response: We revised the figure S5a as this reviewer suggested to show the distribution of all histone modifications across the region upstream of the transcription start site, the gene body, and the region downstream of the transcription termination site.

Referee: 2

Comments to the Author(s)

I enjoyed reading the manuscript. Quiet interesting approach. Overall, very good work. I have some points which may be me not understanding or things to look out for.

Main comments of manuscript:

I think the paper is of great interest and has very good merits to be an outstanding manuscript. It is overall very well written and of great impact! I enjoyed reading it.

I think the most important aspect which I could not understand fully, is the way GEP is calculated. I read the paper Tirosh et al 2006, yet here, the authors use a different formula to calculate GEP. I would very much like to see a better explanation on its formula. What are expected values of GEP for the majority of genes? A gene with 0 plasticity would give you a GEP of 1? In humans, GEPs go from -7 to 1.6... Expected? Also, what about the standard deviation of distribution? Please expand.

From reference 3: Thus, the resulting values have approximately a normal distribution with a mean of 0 and standard deviation of 1, paper: A genetic signature for inter-species variations in gene expression

Response: (1) The term “gene expression plasticity (GEP)” used in this study is equivalent to the term “gene expression responsiveness” described in Tirosh et al 2006. As described in their paper, “A curated data set of yeast expression data comprising more than 1,500 conditions was used to calculate the response of each gene to changing conditions. 'Responsiveness' was defined as the sum of squares of the log₂-ratios over all conditions.” The method used to quantify GEP in this study is very similar to that of Tirosh et al 2006. **The only difference is that instead of sum, we averaged squares of the log₂ fold change over all conditions, and because the values showed a skewed distribution, we further log-transformed them to represent GEP.** As shown in figure 1c, the log-transformed GEP showed normal-like distribution. (2) In Tirosh et al 2006, the authors provided a formula to quantify gene expression divergence between species which is not expression plasticity or responsiveness. (3) According to our GEP quantification method, a gene with a near zero plasticity will result in a large negative value (caused by log-transformation) and genes with a very high plasticity will give a large positive value. (4) To compare the observed GEP to that of expected, we randomly shuffled gene expression changes for each condition to approximate the null distribution. As shown in **figure S1b**, the distribution of observed GEP is much wider than the expected, suggesting genes with high or low plasticity are under regulation.

On the same note, I do not understand how GEP differs from Expression Potential? “While intuition suggests that GEP could be related to expression potential and broadness, our analysis showed that GEP to be poorly correlated with expression level”

Response: “Expression potential” used in the manuscript is equivalent to average gene expression level across conditions. To eliminate the ambiguity, we have changed “expression potential” to “expression level” throughout the manuscript.

Minor comments:

INTRODUCTION section:

“Gene expression plasticity has important implications for organismal fitness.”

- please place reference or expand on concept of plasticity

Response: We have now cited a reference and expanded the concept.

Reference cited: Lopez-Maury, L., Marguerat, S., Bahler, J. 2008 Tuning gene expression to changing environments: from rapid responses to evolutionary adaptation. *Nature reviews. Genetics.* 9, 583-593.

Page-2: “Gene expression plasticity has important implications for organismal fitness [2]. For example, the ability of genes to rapidly tune their expression levels to accommodate changing conditions (such as stress) is crucial for the organism to adapt a new environment hence increasing the fitness.”

RESULTS section:

“Finally, our measurement of GEP is based on direct comparison of gene expression changes before and after treatments/perturbations (challenging conditions) which would in theory more recapitulate the potential of a gene to change its expression as compared to the natural variability of gene expression across tissue or cell types under normal condition.”

- Do not understand the sentence, specifically “more recapitulate”.

Response: Here, we meant to emphasize that including expression changes in response to genetic (e.g. mutants) or environmental perturbations (e.g. stress conditions) would better represent gene expression plasticity. In theory, gene expression changes across normal conditions (such as between different developmental stages, between different cell types, between individuals) could also be used to quantify expression plasticity. To compare the performance of the two approaches, we recalculated GEP using only normal conditions and used genes with high expression plasticity (signal-responsive genes) as benchmark to compare the performance. We found expression plasticity calculated using both normal and challenging conditions (GEP_{N+C}) is moderately correlated with expression plasticity calculated using only normal conditions (GEP_N) (**figure 1d**). However, GEP_{N+C} (used in this study) better recapitulates high expression plasticity of signal-responsive genes as compared to GEP_N (**figure 1e**).

“GEP values of genes in given ontology terms were significantly correlated for all pairwise species comparisons (Figure 2E).”

- not sure how the method for doing this analysis. Is this similar to a gene set enrichment? Or if not, it would be, using the GEP as ranking order.

Response: For each GO term, we first identify all genes with that GO term and used the mean GEP to represent the expression plasticity of a GO term (GEP_{GO}). We then quantified the correlation coefficient of all GEP_{GO} between species to determine whether genes with similar function in one species have concordant expression plasticity in another species.

“Gene body H3K36me3, H3K79me2, and H4K20me1 modifications are associated with restricted expression plasticity”

- How many datasets were used? Enough to do the analysis, over 100?

Response: As shown in figure 6, for this analysis we treated expression plasticity as a constant value and used multiple histone modification datasets (see the table below) to test whether genes with and without a histone modification in the gene body regions show significantly different expression plasticity. Our conclusion was based on (1) majority of samples show consistent difference; (2) the effect is conserved in all four species. As you can see from the table, in vast majority of the case, genes with gene body H3K36me3/H3K79me2/H4K20me1 modifications exhibited significantly lower plasticity. We were unable to collect > 100 histone modification for each species due to limited availability of datasets. However, we believe the relative large number of datasets combined from four species (165 for H3K36me3, 39 for H3K79me2 and 46 for H4K20me1) would allow us to draw a conclusion that the three modifications are associated with low gene expression plasticity.

Modification	Human	Mouse	Fly	Worm	Total
H3K36me3	127 (93%)	20 (85%)	10 (100%)	8 (100%)	165
H3K79me2	20 (81%)	14 (93%)	3 (100%)	2 (100%)	39
H4K20me1	19 (84%)	12 (100%)	9 (100%)	6 (67%)	46

Table shows the number of samples analyzed for each histone modification in each species and the percentage of samples with consistent differences

“gene body regions (Figure S5A), that were strongly correlated with more restricted GEP”

- what is a more restricted GEP? Again, its important to understand the range of GEP and what would be expected.

Response: We meant genes with H3K36me3/H3K79me2/H4K20me1 modifications showed significantly lower expression plasticity as compared to those genes without corresponding modifications. Please see above for our response to the question regarding the range of GEP.

DISCUSSION section:

“there is surprisingly widespread correlation of expression plasticity between orthologs”

- would it be more correct to say, genes with similar GO terms. I believe authors did not perform a homologous analysis, yet similar, I think its GO terms.

Response: we did both analyses. In **figure S2a**, we compared GEP between orthologs between all species and found that, albeit weak, orthologous genes show statistically significant correlation between species. The highest correlation was observed between human and mouse orthologs (Spearman's Rank Correlation, $\rho=0.46$). In **figure 2e**, we compared the correlation of average GEP values of genes with identical GO term between all species and found that the average GEP of genes with similar function tend to be significantly correlated between species.

“Using human Global Run-on Sequencing (GRO-seq)”

- explain briefly what GRO-seq is.

Response: In the revised manuscript we briefly described the meaning and applications of GRO-seq:

Page-17: “Global Run-on Sequencing (GRO-seq) maps the binding sites of transcriptionally active RNA polymerase II and that has been widely used to accurately quantify RNA polymerase II elongation and pause-release. Using human GRO-seq data which...”

CONCLUSIONS section:

“This finding establishes that the changeability of gene expression is an intrinsic gene property with broad biological implications”

- replace “This finding establishes” for “This finding suggest” or similar.... Not sure all in silico evidence is enough to establish. With some transgenic lines carrying motifs, that would establish.

Response: Thanks for pointing this out, we changed the statement to “this finding suggests that ...”

MATERIALS AND METHODS

Quantification of GEP

- as stated before, my main concern.

“We calculated GEP (Table S1) using a similar method as done previously for yeast data, using the following equation:”

- also, I believe the equation is not correct, it seems exponential log.

Response: The equation was incorrectly presented due to a format issue. In the revised manuscript we removed the equation and used text to describe the details of the method in the Results and Methods sections.

“We calculated GEP (**table S1**) using a similar method as done previously for yeast data, first, square of log₂ fold change of mRNA abundance under a given condition as compared to the control was calculated to represent expression change under a condition. Second, expression changes were averaged across all conditions. Third, because the values showed a skewed distribution, we further log₂-transformed them to represent GEP.”

Magnitude of gene expression change in perturbation experiments

“To exclude the possibility that a gene perturbation induced dramatic changes in the gene expression program, which would make comparisons of magnitude meaningless, we only considered datasets where a majority of genes (>60%) were consistently up-regulated or downregulated between conditions in the wild type and mutants”

- Does this mean a essay where 60% genes were up or down-regulated statistically? Replicates were used? Does not make sense.

Response: In this analysis we are comparing the magnitude of gene expression changes across two or more conditions (to proximate GEP) between wild-type and mutant samples. A caveat is that a mutant induces dramatic changes in gene expression pattern so that a comparison of the magnitudes of gene expression between the wild type and mutant is not meaningful. For example, most of the upregulated genes following a treatment become

down-regulated in the mutant samples following the same treatment. To be free of this complexity, we only considered the datasets in which majority (60%) of the differentially expressed genes (between different conditions for a genotype) were shared in the wild type and mutants background. Differential gene expression was identified based on statistic analysis of experimental replicates (T-test, Benjamini–Hochberg corrected $p < 0.01$). We have revised the text for clarity.